# SELFLESS SEQUENTIAL LEARNING

**Rahaf Aljundi**
KU Leuven
ESAT-PSI, Belgium
rahaf.aljundi@gmail.com

**Marcus Rohrbach**
Facebook AI Research
mrf@fb.com

**Tinne Tuytelaars**
KU Leuven
ESAT-PSI, Belgium
tinne.tuytelaars@esat.kuleuven.be

## ABSTRACT

Sequential learning, also called lifelong learning, studies the problem of learning tasks in a sequence with access restricted to only the data of the current task. In this paper we look at a scenario with fixed model capacity, and postulate that the learning process should not be selfish, i.e. it should account for future tasks to be added and thus leave enough capacity for them. To achieve *Selfless Sequential Learning* we study different regularization strategies and activation functions. We find that imposing sparsity at the level of the representation (i.e. neuron activations) is more beneficial for sequential learning than encouraging parameter sparsity. In particular, we propose a novel regularizer, that encourages representation sparsity by means of neural inhibition. It results in few active neurons which in turn leaves more free neurons to be utilized by upcoming tasks. As neural inhibition over an entire layer can be too drastic, especially for complex tasks requiring strong representations, our regularizer only inhibits other neurons in a local neighbourhood, inspired by lateral inhibition processes in the brain. We combine our novel regularizer with state-of-the-art lifelong learning methods that penalize changes to important previously learned parts of the network. We show that our new regularizer leads to increased sparsity which translates in consistent performance improvement on diverse datasets.

## 1 INTRODUCTION

Sequential learning, also referred to as continual, incremental, or lifelong learning (LLL), studies the problem of learning a sequence of tasks, one at a time, without access to the training data of previous or future tasks. When learning a new task, a key challenge in this context is how to avoid catastrophic interference with the tasks learned previously (French, 1999; Li & Hoiem, 2016). Some methods exploit an additional episodic memory to store a small amount of previous tasks data to regularize future task learning (e.g. Lopez-Paz et al. (2017)). Others store previous tasks models and at test time, select one model or merge the models (Rusu et al., 2016; Aljundi et al., 2016; Lee et al., 2017). In contrast, in this work we are interested in the challenging situation of learning a sequence of tasks *without access to any previous or future task data* and *restricted to a fixed model capacity*, as also studied in Kirkpatrick et al. (2016); Aljundi et al. (2017); Fernando et al. (2017); Mallya & Lazebnik (2017); Serrà et al. (2018). This scenario not only has many practical benefits, including privacy and scalability, but also resembles more closely how the mammalian brain learns tasks over time.

The mammalian brain is composed of billions of neurons. Yet at any given time, information is represented by only a few active neurons resulting in a sparsity of 90-95% (Lennie, 2003). In neural biology, *lateral inhibition* describes the process where an activated neuron reduces the activity of its weaker neighbors. This creates a powerful decorrelated and compact representation with minimum interference between different input patterns in the brain (Yu et al., 2014). This is in stark contrast with artificial neural networks, which typically learn dense representations that are highly entangled (Bengio et al., 2009). Such an entangled representation is quite sensitive to changes in the

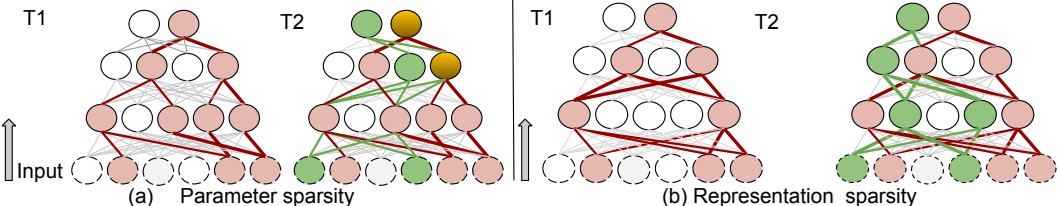

Figure 1: The difference between parameter sparsity (a) and representation sparsity (b) in a simple two tasks case. First layer indicates input patterns. Learning the first task utilizes parts indicated in red. Task 2 has different input patterns and uses parts shown in green. Orange indicates changed neurons activations as a result of the second task. In (a), when an example from the first task is encountered again, the activations of the first layer will not be affected by the changes, however, the second and later layer activations are changed. Such interference is largely reduced when imposing sparsity on the representation (b).

input patterns, in that it responds differently to input patterns with only small variations. French (1999) suggests that an overlapped internal representation plays a crucial role in catastrophic forgetting and reducing this overlap would result in a reduced interference. Cogswell et al. (2015) show that when the amount of overfitting in a neural network is reduced, the representation correlation is also reduced. As such, learning a disentangled representation is more powerful and less vulnerable to catastrophic interference. However, if the learned disentangled representation at a given task is not sparse, only little capacity is left for the learning of new tasks. This would in turn result in either an underfitting to the new tasks or again a forgetting of previous tasks. In contrast, a sparse and decorrelated representation would lead to a powerful representation and at the same time enough free neurons that can be changed without interference with the neural activations learned for the previous tasks.

In general, sparsity in neural networks can be thought of either in terms of the network parameters or in terms of the representation (i.e., the activations). In this paper we postulate, and confirm experimentally, that a sparse and decorrelated representation is preferable over parameter sparsity in a sequential learning scenario. There are two arguments for this: first, a sparse representation is less sensitive to new and different patterns (such as data from new tasks) and second, the training procedure of the new tasks can use the free neurons leading to less interference with the previous tasks, hence reducing forgetting. In contrast, when the effective parameters are spread among different neurons, changing the ineffective ones would change the function of their corresponding neurons and hence interfere with previous tasks (see also Figure 1). Based on these observations, we propose a new regularizer that exhibits a behavior similar to the lateral inhibition in biological neurons. The main idea of our regularizer is to penalize neurons that are active at the same time. This leads to more sparsity and a decorrelated representation. However, complex tasks may actually require multiple active neurons in a layer at the same time to learn a strong representation. Therefore, our regularizer, **Sparse coding through Local Neural Inhibition and Discounting (`SLNID`)**, only penalizes neurons locally. Furthermore, we don't want inhibition to affect previously learned tasks, even if later tasks use neurons from earlier tasks. An important component of `SLNID` is thus to discount inhibition from/to neurons which have high *neuron importance* – a new concept that we introduce in analogy to parameter importance (Kirkpatrick et al., 2016; Zenke et al., 2017; Aljundi et al., 2017). When combined with a state-of-the-art important parameters preservation method (Aljundi et al., 2017; Kirkpatrick et al., 2016), our proposed regularizer leads to sparse and decorrelated representations which improves the lifelong learning performance.

Our contribution is threefold. First, we direct attention to **Selfless Sequential Learning** and study a diverse set of representation based regularizers, parameter based regularizers, as well as sparsity inducing activation functions to this end. These have not been studied extensively in the lifelong learning literature before. Second, we propose a novel regularizer, `SLNID`, which is inspired by lateral inhibition in the brain. Third, we show that our proposed regularizer consistently outperforms alternatives on three diverse datasets (Permuted MNIST, CIFAR, Tiny Imagenet) and we compare to and outperform state-of-the-art LLL approaches on an 8-task object classification challenge. `SLNID` can be applied to different regularization based LLL approaches, and we show experiments with MAS (Aljundi et al., 2017) and EWC (Kirkpatrick et al., 2016).

In the following, we first discuss related approaches to LLL and different regularization criteria from a LLL perspective (Section 2). We proceed by introducing Selfless Sequential Learning and detailing our novel regularizer (Section 3). Section 4 describes our experimental evaluation, while Section 5 concludes the paper.

## 2 RELATED WORK

The goal in lifelong learning is to learn a sequence of tasks without catastrophic forgetting of previously learned ones (Thrun & Mitchell, 1995). One can identify different approaches to introducing lifelong learning in neural networks. Here, we focus on learning a sequence of tasks using a fixed model capacity, i.e. with a fixed architecture and fixed number of parameters. Under this setting, methods either follow a pseudo rehearsal approach, i.e. using the new task data to approximate the performance of the previous task (Li & Hoiem, 2016; Triki et al., 2017), or aim at identifying the important parameters used by the current set of tasks and penalizing changes to those parameters by new tasks (Kirkpatrick et al., 2016; Zenke et al., 2017; Aljundi et al., 2017; Chaudhry et al., 2018; Liu et al., 2018). To identify the important parameters for a given task, Elastic Weight Consolidation (Kirkpatrick et al., 2016) uses an approximation of the Fisher information matrix computed after training a given task. Liu et al. (2018) suggest a network reparameterization to obtain a better diagonal approximation of the Fisher Information matrix of the network parameters. Path Integral (Zenke et al., 2017) estimates the importance of the network parameters while learning a given task by accumulating the contribution of each parameter to the change in the loss. Chaudhry et al. (2018) suggest a KL-divergence based generalization of Elastic Weight Consolidation and Path Integral. Memory Aware Synapses (Aljundi et al., 2017) estimates the importance of the parameters in an online manner without supervision by measuring the sensitivity of the learned function to small perturbations on the parameters. This method is less sensitive to the data distribution shift, and a local version proposed by the authors resembles applying Hebb rule (Hebb, 2002) to consolidate the important parameters, making it more biologically plausible.

A common drawback of all the above methods is that learning a task could utilize a good portion of the network capacity, leaving few "free" neurons to be adapted by the new task. This in turn leads to inferior performance on the newly learned tasks or forgetting the previously learned ones, as we will show in the experiments. Hence, we study the role of sparsity and representation decorrelation in sequential learning. This aspect has not received much attention in the literature yet. Very recently, (Serrà et al., 2018) proposed to overcome catastrophic forgetting through learned hard attention masks for each task with L1 regularization imposed on the accumulated hard attention masks. This comes closer to our approach although we study and propose a regularization scheme on the learned representation.

The concept of reducing the representation overlap has been suggested before in early attempts towards overcoming catastrophic forgetting in neural networks (French, 1999). This has led to several methods with the goal of orthogonalizing the activations (French, 1992; 1994; Kruschke, 1992; 1993; Sloman & Rumelhart, 1992). However, these approaches are mainly designed for specific architectures and activation functions, which makes it hard to integrate them in recent neural network structures.

The sparsification of neural networks has mostly been studied for compression. SVD decomposition can be applied to reduce the number of effective parameters (Xue et al., 2013). However, there is no guarantee that the training procedure converges to a low rank weight matrix. Other works iterate between pruning and retraining of a neural network as a post processing step (Liu et al., 2015; Sun et al., 2016; Aghasi et al., 2017; Louizos et al., 2017). While compressing a neural network by removing parameters leads to a sparser neural network, this does not necessarily lead to a sparser representation. Indeed, a weight vector can be highly sparse but spread among the different neurons. This reduces the effective size of a neural network, from a compression point of view, but it would not be beneficial for later tasks as most of the neurons are already occupied by the current set of tasks. In our experiments, we show the difference between using a sparse penalty on the representation versus applying it to the weights.

## 3 SELFLESS SEQUENTIAL LEARNING

One of the main challenges in single model sequential learning is to have capacity to learn new tasks and at the same time avoid catastrophic forgetting of previous tasks as a result of learning new tasks. In order to prevent catastrophic forgetting, importance weight based methods such as EWC (Kirkpatrick et al., 2016) or MAS (Aljundi et al., 2017) introduce an importance weight $\Omega_k$ for each parameter $\theta_k$ in the network. While these methods differ in how to estimate the important parameters,

all of them penalize changes to important parameters when learning a new task $T_n$ using $L_2$ penalty:

$$T_n: \quad \min_\theta \frac{1}{M} \sum_{m=1}^M \mathcal{L}(y_m, f(x_m, \theta^n)) + \lambda_\Omega \sum_k \Omega_k (\theta_k^n - \theta_k^{n-1})^2 \tag{1}$$

where $\theta^{n-1} = \{\theta_k^{n-1}\}$ are the optimal parameters learned so far, i.e. before the current task. $\{x_m\}$ is the set of $M$ training inputs, with $\{f(x_m, \theta^n)\}$ and $\{y_m\}$ the corresponding predicted and desired outputs, respectively. $\lambda_\Omega$ is a trade-off parameter between the new task objective $\mathcal{L}$ and the changes on the important parameters, i.e. the amount of forgetting.

In this work we introduce an additional regularizer $R_{\text{SSL}}$ which encourages sparsity in the activations $H_l = \{h_i^m\}$ for each layer $l$.

$$T_n: \quad \min_\theta \frac{1}{M} \sum_{m=1}^M \mathcal{L}(y_m, f(x_m, \theta^n)) + \lambda_\Omega \sum_k \Omega_k (\theta_k^n - \theta_k^{n-1})^2 + \lambda_{\text{SSL}} \sum_l R_{\text{SSL}}(H_l) \tag{2}$$

$\lambda_{\text{SSL}}$ and $\lambda_\Omega$ are trade-off parameters that control the contribution of each term. When training the first task ($n = 1$), $\Omega_k = 0$.

## 3.1 SPARSE CODING THROUGH NEURAL INHIBITION (SNI)

Now we describe how we obtain a sparse and decorrelated representation. In the literature sparsity has been proposed by Glorot et al. (2011) to be combined with the rectifier activation function (ReLU) to control unbounded activations and to increase sparsity. They minimize the $L_1$ norm of the activations (since minimizing the $L_0$ norm is an NP hard problem). However, $L_1$ norm imposes an equal penalty on all the active neurons leading to small activation magnitude across the network.

Learning a decorrelated representation has been explored before with the goal of reducing overfitting. This is usually done by minimizing the Frobenius norm of the covariance matrix corrected by the diagonal, as in Cogswell et al. (2015) or Xiong et al. (2016). Such a penalty results in a decorrelated representation but with activations that are mostly close to a non zero mean value. We merge the two objectives of sparse and decorrelated representation resulting in the following objective:

$$R_{\text{SNI}}(H_l) = \frac{1}{M} \sum_{i,j} \sum_m h_i^m h_j^m, \quad i \neq j \tag{3}$$

where we consider a hidden layer $l$ with activations $H_l = \{h_i^m\}$ for a set of inputs $X = \{x_m\}$ and $i, j \in 1, .., N$ running over all $N$ neurons in the hidden layer. This formula differs from minimizing the Frobenius norm of the covariance matrix in two simple yet important aspects:

(1) In the case of a ReLU activation function, used in most modern architectures, a neuron is active if its output is larger than zero, and zero otherwise. By assuming a close to zero mean of the activations, $\mu_i \simeq 0 \, \forall i \in 1, .., N$, we minimize the correlation between any two active neurons.

(2) By evaluating the derivative of the presented regularizer w.r.t. the activation, we get:

$$\frac{\partial R_{\text{SNI}}(H_l)}{\partial h_i^m} = \frac{1}{M} \sum_{j \neq i} h_j^m \tag{4}$$

i.e., each active neuron receives a penalty from every other active neuron that corresponds to that other neuron's activation magnitude. In other words, if a neuron fires, with a high activation value, for a given example, it will suppress firing of other neurons for that same example. Hence, this results in a decorrelated sparse representation.

## 3.2 SPARSE CODING THROUGH LOCAL NEURAL INHIBITION (SLNI)

The loss imposed by the SNI objective will only be zero when there is at most one active neuron per example. This seems to be too harsh for complex tasks that need a richer representation. Thus, we suggest to relax the objective by imposing a spatial weighting to the correlation penalty. In other words, an active neuron penalizes mostly its close neighbours and this effect vanishes for neurons further away. Instead of uniformly penalizing all the correlated neurons, we weight the correlation penalty between two neurons with locations $i$ and $j$ using a Gaussian weighting. This gives

$$R_{\text{SLNI}}(H_l) = \frac{1}{M} \sum_{i,j} e^{-\frac{(i-j)^2}{2\sigma^2}} \sum_m h_i^m h_j^m, \quad i \neq j \tag{5}$$

As such, each active neuron inhibits its neighbours, introducing a locality in the network inspired by biological neurons. While the notion of neighbouring neurons is not well established in a fully connected network, our aim is to allow few neurons to be active and not only one, thus those few activations don't have to be small to compensate for the penalty. $\sigma^2$ is a hyper parameter representing the scale at which neurons can affect each other. Note that this is somewhat more flexible than decorrelating neurons in fixed groups as used in Xiong et al. (2016). Our regularizer inhibits locally the active neurons leading to a sparse coding through local neural inhibition.

### 3.3 Neuron importance for discounting inhibition

Our regularizer is to be applied for each task in the learning sequence. In the case of tasks with completely different input patterns, the active neurons of the previous tasks will not be activated given the new tasks input patterns. However, when the new tasks are of similar or shared patterns, neurons used for previous tasks will be active. In that case, our penalty would discourage other neurons from being active and encourage the new task to adapt the already active neurons instead. This would interfere with the previous tasks and could increase forgetting which is exactly what we want to overcome. To avoid such interference, we add a weight factor taking into account the importance of the neurons with respect to the previous tasks. To estimate the importance of the neurons, we use as a measure the sensitivity of the loss at the end of the training to their changes. This is approximated by the gradients of the loss w.r.t. the neurons outputs (before the activation function) evaluated at each data point. To get an importance value, we then accumulate the absolute value of the gradients over the given data points obtaining importance weight $\alpha_i$ for neuron $n_i$:

$$\alpha_i = \frac{1}{M} \sum_{m=1}^{M} \mid g_i(x_m) \mid \quad , \quad g_i(x_m) = \frac{\partial(\mathcal{L}(y_m, f(x_m, \theta^n)))}{\partial n_i^m} \tag{6}$$

where $n_i^m$ is the output of neuron $n_i$ for a given input example $x_m$, and $\theta^n$ are the parameters after learning task $n$. This is in line with the estimation of the parameters importance in Kirkpatrick et al. (2016) but considering the derivation variables to be the neurons outputs instead of the parameters. Instead of relying on the gradient of the loss, we can also use the gradient of the learned function, i.e. the output layer, as done in Aljundi et al. (2017) for estimating the parameters importance. During the early phases of this work, we experimented with both and observed a similar behaviour. For sake of consistency and computational efficiency we utilize the gradient of the function when using Aljundi et al. (2017) as LLL method and the gradient of the loss when experimenting with EWC (Kirkpatrick et al., 2016). Then, we can weight our regularizer as follows:

$$R_{\text{SLNID}}(H_l) = \frac{1}{M} \sum_{i,j} e^{-(\alpha_i+\alpha_j)} e^{-\frac{(i-j)^2}{2\sigma^2}} \sum_m h_i^m h_j^m, \quad i \neq j \tag{7}$$

which can be read as: if an important neuron for a previous task is active given an input pattern from the current task, it will not suppress the other neurons from being active neither be affected by other active neurons. For all other active neurons, local inhibition is deployed. The final objective for training is given in Eq. 2, setting $R_{SSL} := R_{\text{SLNID}}$ and $\lambda_{\text{SSL}} := \lambda_{\text{SLNID}}$. We refer to our full method as Sparse coding through Local Neural Inhibition and Discounting (SLNID).

## 4 Experiments

In this section we study the role of standard regularization techniques with a focus on sparsity and decorrelation of the representation in a sequential learning scenario. We first compare different activation functions and regularization techniques, including our proposed SLNID, on permuted MNIST (Sec. 4.1). Then, we compare the top competing techniques and our proposed method in the case of sequentially learning CIFAR-100 classes and Tiny Imagenet classes (Sec. 4.2). Our SLNID regularizer can be integrated in any importance weight-based lifelong learning approach such as (Kirkpatrick et al., 2016; Zenke et al., 2017; Aljundi et al., 2017). Here we focus on Memory Aware Synapses (Aljundi et al., 2017) (MAS), which is easy to integrate and experiment with and has shown superior performance (Aljundi et al., 2017). However, we also show results with Elastic weight consolidation (Kirkpatrick et al., 2016)(EWC) in Sec. 4.3. Further, we ablate the components of our regularizer, both in the standard setting (Sec. 4.4) as in a setting without hard task boundaries (Sec. 4.5). Finally, we show how our regularizer improves the state-of-the-art performance on a sequence of object recognition tasks (Sec. 4.6).

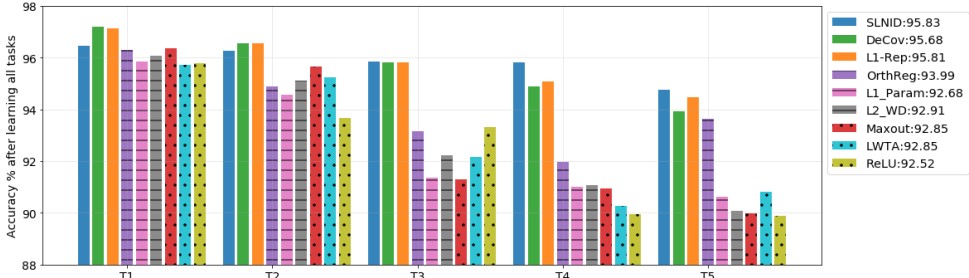

Figure 2: Comparison of different regularization techniques on 5 permuted MNIST sequence. Representation based regularizers are solid bars, bars with lines represent parameters regularizers, dotted bars represent activation functions. Average test accuracy over all tasks is given in the legend. Representation based regularizers achieve higher performance than other compared methods including parameters based regularizers. Our regularizer, `SLNID`, performs the best on the last two tasks indicating that more capacity is left to learn these tasks.

## 4.1 AN IN-DEPTH COMPARISON OF REGULARIZERS AND ACTIVATION FUNCTIONS FOR SELFLESS SEQUENTIAL LEARNING

We study possible regularization techniques that could lead to less interference between the different tasks in a sequential learning scenario either by enforcing sparsity or decorrelation. Additionally, we examine the use of activation functions that are inspired by lateral inhibition in biological neurons that could be advantageous in sequential learning. MAS Aljundi et al. (2017) is used in all cases as LLL method.

**Representation Based methods:**

- `L1-Rep`: To promote representational sparsity, an $L_1$ penalty on the activations is used.

- `Decov` (Cogswell et al., 2015) aims at reducing overfitting by decorrelating neuron activations. To do so, it minimizes the Frobenius norm of the covariance matrix computed on the activations of the current batch after subtracting the diagonal to avoid penalizing independent neuron activations.

**Activation functions:**

- `Maxout` network (Goodfellow et al., 2013b) utilizes the maxout activation function. For each group of neurons, based on a fixed window size, only the maximum activation is forwarded to the next layer. The activation function guarantees a minimum sparsity rate defined by the window size.

- `LWTA` (Srivastava et al., 2013): similar idea to the Maxout network except that the non-maximum activations are set to zero while maintaining their connections. In contrast to Maxout, LWTA keeps the connections of the inactive neurons which can be occupied later once they are activated without changing the previously active neuron connections.

- `ReLU` (Glorot et al., 2011) The rectifier activation function (ReLU) used as a baseline here and indicated in later experiments as `No-Reg` as it represents the standard setting of sequential learning on networks with ReLU. All the studied regularizers use ReLU as activation function.

**Parameters based regularizers:**

- `OrthReg` (Rodríguez et al., 2016): Regularizing CNNs with locally constrained decorrelations. It aims at decorrelating the feature detectors by minimizing the cosine of the angle between the weight vectors resulting eventually in orthogonal weight vectors.

- `L2-WD`: Weight decay with $L_2$ norm (Krogh & Hertz, 1992) controls the complexity of the learned function by minimizing the magnitude of the weights.

- `L1-Param`: $L_1$ penalty on the parameters to encourage a solution with sparse parameters.

Dropout is not considered as its role contradicts our goal. While dropout can improve each task performance and reduce overfitting, it acts as a model averaging technique. By randomly masking neurons, dropout forces the different neurons to work independently. As such it encourages a redundant representation. As shown by (Goodfellow et al., 2013a) the best network size for classifying MNIST digits when using dropout was about $50\%$ more than without it. Dropout steers the learning of a task towards occupying a good portion of the network capacity, if not all of it, which contradicts the sequential learning needs.

**Experimental setup.** We use the MNIST dataset (LeCun et al., 1998) as a first task in a sequence of 5 tasks, where we randomly permute all the input pixels differently for tasks 2 to 5. The goal is to classify MNIST digits from all the different permutations. The complete random permutation of the pixels in each task requires the neural network to instantiate a new neural representation for each pattern. A similar setup has been used by Kirkpatrick et al. (2016); Zenke et al. (2017); Goodfellow et al. (2013a) with different percentage of permutations or different number of tasks.

As a base network, we employ a multi layer perceptron with two hidden layers and a Softmax loss.

Figure 3: Comparison of different regularization techniques on a sequence of ten tasks from (a) Cifar split and (b) Tiny ImageNet split. The legend shows average test accuracy over all tasks. Simple L1-norm regularizer (L1-Rep) doesn't help in such more complex tasks. Our regularizer SLNID achieves an improvement of 2% over Decov and $4 - 8\%$ compared to No-Reg.

We experiment with different number of neurons in the hidden layers $\{128, 64\}$. For SLNID we evaluate the effect of $\lambda_{\text{SLNID}}$ on the performance and the obtained sparsity in Figure 4. In general, the best $\lambda_{\text{SLNID}}$ is the minimum value that maintains similar or better accuracy on the first task compared to the unregularized case, and we suggest to use this as a rule-of-thumb to set $\lambda_{\text{SLNID}}$. For $\lambda_\Omega$, we have used a high $\lambda_\Omega$ value that ensures the least forgetting which allows us to test the effect on the later tasks performance. Note that better average accuracies can be obtained with tuned $\lambda_\Omega$. Please refer to Appendix A for hyperparameters and other details.

**Results:** Figure 2 presents the test accuracy on each task at the end of the sequence, achieved by the different regularizers and activation functions on the network with hidden layer of size 128. Results on a network with hidden layer size 64 are shown in the Appendix B. Clearly, in all the different tasks, the representational regularizers show a superior performance to the other studied techniques. For the regularizers applied to the parameters, L2-WD and L1-Param do not exhibit a clear trend and do not systematically show an improvement over the use of the different activation functions only. While OrthReg shows a consistently good performance, it is lower than what can be achieved by the representational regularizers. It is worth noting the L1-Rep yields superior performance over L1-Param. This observation is consistent across different sizes of the hidden layers (in Appendix B) and shows the advantage of encouraging sparsity in the activations compared to that in the parameters. Regarding the activation functions, Maxout and LWTA achieve a slightly higher performance than ReLU. We did not observe a significant difference between the two activation functions. However, the improvement over ReLU is only moderate and does not justify the use of a fixed window size and special architecture design. Our proposed regularizer SLNID achieves high if not the highest performance in all the tasks and succeeds in having a stable performance. This indicates the ability of SLNID to direct the learning process towards using minimum amount of neurons and hence more flexibility for upcoming tasks.

**Representation sparsity & important parameter sparsity.** Here we want to examine the effect of our regularizer on the percentage of parameters that are utilized after each task and hence the capacity left for the later tasks. On the network with hidden layer size 128, we compute the percentage of parameters with $\Omega_k < 10^{-2}$, with $\Omega_k$, see Appendix A, the importance weight multiplier estimated and accumulated over tasks. Those parameters can be seen as unimportant and "free" for later tasks. Figure 4(top) shows the percentage of the unimportant (free) parameters in the first layer after each task for different $\lambda_{\text{SLNID}}$ values along with the achieved average test accuracy at the end of the sequence. It is clear that the larger $\lambda_{\text{SLNID}}$, i.e., the more neural inhibition, the smaller the percentage of important parameters. Apart from the highest $\lambda_{\text{SLNID}}$ where tasks couldn't reach their top performance due to too strong inhibition, improvement over the No-Reg is always observed. The optimal value for lambda seems to be the one that remains close to the optimal performance on the current task, while utilizing the minimum capacity feasible. Next, we compute the average activation per neuron, in the first layer, over all the examples and plot the corresponding histogram for SLNID, DeCov, L1-Rep, L1-Param and No-Reg in Figure 4(bottom) at their setting that yielded the results shown in Figure 2. SLNID has a peak at zero indicating representation sparsity while the other methods values are spread along the line. This seems to hint at the effectiveness of our approach SLNID in learning a sparse yet powerful representation and in turn in a minimal interference between tasks.

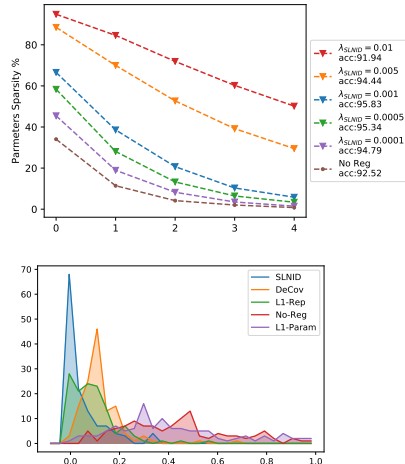

Figure 4: On the 5 permuted MNIST sequence, hidden layer=128, Top: percentage of unused parameters in the 1st layer using different $\lambda_{\text{SLNID}}$; Bottom: histogram of neural activations on the first task.

| | Permuted mnist | | Cifar | |
|---|---|---|---|---|
| h-layer dim. | 128 | 64 | 256 | 128 |
| No-Reg | 92.67 | 90.72 | 55.06 | 55.3 |
| SNI | 95.79 | **94.89** | 55.30 | 55.75 |
| SNID | 95.90 | 93.82 | 61.00 | 60.90 |
| SLNI | **95.95** | 94.87 | 56.06 | 55.79 |
| SLNID | 95.83 | 93.89 | **63.30** | **61.16** |
| Multi-Task Joint Training* | 97.30 | 96.80 | 70.99 | 71.95 |

Table 1: SLNID ablation. Average test accuracy per task after training the last task in %. * denotes that Multi-Task Joint Training violates the LLL scenario as it has access to all tasks at once and thus can be seen as an upper bound.

| Method | Avg-acc |
|---|---|
| Finetune | 32.67 |
| LWF (Li & Hoiem, 2016) | 49.49 |
| EBLL (Triki et al., 2017) | 50.29 |
| IMM (Lee et al., 2017) | 43.4 |
| Path Integral (Zenke et al., 2017) | 50.49 |
| EWC (Kirkpatrick et al., 2016) | 50.00 |
| MAS (Aljundi et al., 2017) | 52.69 |
| SLNID-fc Pretrained (ours) | 53.77 |
| SLNID-fc randomly initialized (ours) | **54.50** |

Table 2: 8 tasks object recognition sequence. Average test accuracy per task after training the last task in %.

## 4.2 10 Task sequences on CIFAR-100 and Tiny Imagenet

While the previous section focused on learning a sequence of tasks with completely different input patterns and same objective, we now study the case of learning different categories of one dataset. For this we split the CIFAR-100 and the Tiny ImageNet (Yao & Miller, 2015) dataset into ten tasks, respectively. We have 10 and 20 categories per task for CIFAR-100 and Tiny ImagNet, respectively. Further details about the experimental setup can be found in appendix A.

We compare the top competing methods from the previous experiments, L1-Rep, DeCov and our SLNID, and No-Reg as a baseline, ReLU in previous experiment. Similarly, MAS Aljundi et al. (2017) is used in all cases as LLL method. Figures 3(a) and 3(b) show the performance on each of the ten tasks at the end of the sequence. For both datasets, we observe that our SLNID performs overall best. L1-Rep and DeCov continue to improve over the non regularized case No-Reg. These results confirm our proposal on the importance of sparsity and decorrelation in sequential learning.

## 4.3 SLNID with EWC (Kirkpatrick et al., 2016)

We have shown that our proposed regularizer SLNID exhibits stable and superior performance on the different tested networks when using MAS as importance weight preservation method. To prove the effectiveness of our regularizer regardless of the used importance weight based method, we have tested SLNID on the 5 tasks permuted MNIST sequence in combination with Elastic Weight Consolidation (EWC,Kirkpatrick et al. (2016)) and obtained a boost in the average performance at the end of the learned sequence equal to $3.1\%$ on the network with hidden layer size 128 and a boost of $2.8\%$ with hidden layer size 64. Detailed accuracies are shown in Appendix B. It is worth noting that with both MAS and EWC our SLNID was able obtain better accuracy using a network with a 64-dimensional hidden size than when training without regularization No-Reg on a network of double that size (128), indicating that SLNID allows to use neurons much more efficiently.

## 4.4 Ablation Study

Our method can be seen as composed of three components: the neural inhibition, the locality relaxation and the neuron importance integration. To study how these components perform individually, Table 1 reports the average accuracy at the end of the Cifar 100 and permuted MNIST sequences for each variant, namely, SNID without neuron importance (SNI), SNID, SLNID without neuron importance (SLNI) in addition to our full SLNID regularizer. As we explained in Section 3, when tasks have completely different input patterns, the neurons that were activated on the previous task examples will not fire for new task samples and exclusion of important neurons is not mandatory. However, when sharing is present between the different tasks, a term to prevent SLNID from causing any interference is required. This is manifested in the reported results: for permuted MNIST, all the variants work nicely alone, as a result of the simplicity and the disjoint nature of this sequence. However, in the Cifar 100 sequence, the integration of the neuron importance in the SNID and SLNID regularizers exclude important neurons from the inhibition, resulting in a clearly better performance. The locality in SLNID improves the performance in the Cifar sequence, which suggests that a richer representation is needed and multiple active neurons should be tolerated.

## 4.5 SEQUENTIAL LEARNING WITHOUT HARD TASK BOUNDARIES

In the previous experiments, we considered the standard task based scenario as in (Li & Hoiem, 2016; Zenke et al., 2017; Aljundi et al., 2017; Serrà et al., 2018), where at each time step we receive a task along with its training data and a new classification layer is initiated for the new task, if needed. Here, we are interested in a more realistic scenario where the data distribution shifts gradually without hard task boundaries.

To test this setting, we use the Cifar 100 dataset. Instead of considering a set of 10 disjoint tasks each composed of 10 classes, as in the previous experiment (Sec. 4.2), we now start by sampling with high probability $(2/3)$ from the first 10 classes and with low probability $(1/3)$ from the rest of the classes. We train the network (same architecture as in Sec. 4.2) for a few epochs and then change the sampling probabilities to be high $(2/3)$ for classes $11 - 20$ and low $(1/3)$ for the remaining classes. This process is repeated until sampling with high probability from the last 10 classes and low from the rest. We use one shared classification layer throughout and estimate the importance weights and the neurons importance after each training step (before changing the sampling probabilities). We consider 6 variants: our `SLNID`, the ablations `SLNI` and without regularizer `No-Reg`, as in Section 4.4, as well each of these three trained without the MAS importance weight regularizer of Aljundi et al. (2017), denoted as `w/o MAS`. Table 3 presents the accuracy averaged over the ten groups of ten

| Method | Avg.acc -tasks models |
|---|---|
| `No-Reg w/o MAS` | 69.20% |
| `SLNI w/o MAS` | 72.14% |
| `SLNID w/o MAS` | 73.03% |
| `No-Reg` | 66.88% |
| `SLNI` | 71.32% |
| `SLNID` | 72.33% |

| Method | Avg.acc-last model |
|---|---|
| `No-Reg w/o MAS` | 65.15% |
| `SLNI w/o MAS` | 63.54% |
| `SLNID w/o MAS` | 70.75% |
| `No-Reg` | 66.33% |
| `SLNI` | 64.50% |
| `SLNID` | 70.94% |

Table 3: No tasks boundaries test case on Cifar 100. Top block, avg. acc on each group of classes using each group model. Bottom block, avg. acc. on each group at the end of the training.

classes, using each group model (i.e. the model trained when this group was sampled with high probability) in the top block and the average accuracy on each of the ten groups at the end of the training (middle and bottom block). We can deduce the following: 1) `SLNID` improves the performance considerably (by more than 4%) even without importance weight regularizer. 2) In this scenario without hard task boundaries there is less forgetting than in the scenario with hard task boundaries studied in Section 4.2 for Cifar (difference between rows in top block to corresponding rows in middle block). As a result, the improvement obtained by deploying the importance weight regularizer is moderate: at 70.75%, `SLNID w/o MAS` is already better than `No-Reg` reaching 66.33%. 3) While `SLNI` without MAS improves the individual models performance (72.14% compared to 69.20%), it fails to improve the overall performance at the end of the sequence (63.54% compared to 65.15%), as important neurons are not excluded from the penalty and hence they are changed or inhibited leading to tasks interference and performance deterioration.

## 4.6 COMPARISON WITH THE STATE OF THE ART

To compare our proposed approach with the different state-of-the-art sequential learning methods, we use a sequence of 8 different object recognition tasks, introduced in Aljundi et al. (2017). The sequence starts from AlexNet (Krizhevsky et al., 2012) pretrained on ImageNet (Russakovsky et al., 2015) as a base network, following the setting of Aljundi et al. (2017). More details are in Appendix A.4. We compare against the following: *Learning without Forgetting* (Li & Hoiem, 2016) (`LwF`), *Incremental Moment Matching* (Lee et al., 2017) (`IMM`), *Path Integral* (Zenke et al., 2017) and sequential finetuning (`FineTuning`), in addition to the case of `MAS` (Aljundi et al., 2017) alone, i.e. our `No-Reg` before. Compared methods were run with the exact same setup as in Aljundi et al. (2017). For our regularizer, we disable dropout, since dropout encourages redundant activations which contradicts our regularizer's role. Also, since the network is pretrained, the locality introduced in `SLNID` may conflict with the already pretrained activations. For this reason, we also test `SLNID` with randomly initialized fully connected layers. Our regularizer is applied with `MAS` as a sequential learning method. Table 2 reports the average test accuracy at the end of the sequence achieved by each method. `SLNID` improves even when starting from a pretrained network and disabling dropout. Surprisingly, even with randomly initialized fully connected layers, `SLNID` improves 1.8% over the state of the art using a fully pretrained network.

## 5 CONCLUSION

In this paper we study the problem of sequential learning using a network with fixed capacity – a prerequisite for a scalable and computationally efficient solution. A key insight of our approach is that in the context of sequential learning (as opposed to other contexts where sparsity is imposed, such as network compression or avoiding overfitting), sparsity should be imposed at the level of the representation rather than at the level of the network parameters. Inspired by lateral inhibition in the mammalian brain, we impose sparsity by means of a new regularizer that decorrelates nearby active neurons. We integrate this in a model which *learns selflessly* a new task by leaving capacity for future tasks and at the same time *avoids forgetting* previous tasks by taking into account neurons importance.

**Acknowledgment:** The first author's PhD is funded by an FWO scholarship.

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

APPENDIX

## A   DETAILS ON THE EXPERIMENTAL SETUP

In all designed experiments, our regularizer is applied to the neurons of the fully connected layers. As a future work, we plan to integrate it in the convolutional layers.

### A.1   PERMUTED MNIST

The used network is composed of two fully connected layers. All tasks are trained for 10 epochs with a learning rate $10^{-2}$ using SGD optimizer. ReLU is used as an activation function unless mentioned otherwise. Throughout the experiment, we used a scale $\sigma$ for the Gaussian function used for the local inhibition equal to $1/6$ of the hidden layer size. For all competing regularizers, we tested different hyper parameters from $10^{-2}$ to $10^{-9}$ and report the best one. For $\lambda_\Omega$, we have used a high $\lambda_\Omega$ value that ensures the least forgetting. This allows us to examine the degradation in the performance on the later tasks compared to those learned previously as a result of lacking capacity. Note that better average accuracies can be obtained with tuned $\lambda_\Omega$.

In section 4.1 we estimated the free capacity in the network with the percentage of $\Omega_k < 10^{-2}$, with $\Omega_k$, the importance weight multiplier estimated and accumulated over tasks. We consider $\Omega_k < 10^{-2}$ of negligible importanc as in a network trained without a sparsity regularizer, $\Omega_{ij} < 10^{-2}$ covers the first 10 percentiles.

### A.2   CIFAR-100

As a base network, we use a network similar to the one used by Zenke et al. (2017) but without dropout. We evaluate two variants with hidden size $N = \{256, 128\}$. Throughout the experiment, we again used a scale $\sigma$ for the Gaussian function equal to $1/6$ of the hidden layer size. We train the different tasks for 50 epochs with a learning rate of $10^{-2}$ using SGD optimizer.

### A.3   TINY IMAGENET

We split the Tiny ImageNet dataset (Yao & Miller, 2015) into ten tasks, each containing twenty categories to be learned at once. As a base network, we use a variant of VGG (Simonyan & Zisserman, 2014). For architecture details, please refer to Table 4 below.

| Layer | # filters/neurons |
|---|---|
| Convolution | 64 |
| Max Pooling | - |
| Convolution | 128 |
| Max Pooling | - |
| Convolution | 256 |
| Max Pooling | - |
| Convolution | 256 |
| Max Pooling | - |
| Convolution | 512 |
| Convolution | 512 |
| Fully connected | 500 |
| Fully connected | 500 |
| Fully connected | 20 |

Table 4: Architecture of the network used in the Tiny Imagenet experiment.

Throughout the experiment, we again used a scale $\sigma$ for the Gaussian function equal to $1/6$ of the hidden layer size.

### A.4   8 TASK OBJECT RECOGNITION SEQUENCE

The 8 tasks sequence is composed of: 1. Oxford *Flowers* (Nilsback & Zisserman, 2008), 2. MIT *Scenes* (Quattoni & Torralba, 2009), 3. Caltech-UCSD *Birds* (Welinder et al., 2010), 4. Stanford `Cars` (Krause et al., 2013); 5. FGVC-`Aircraft` (Maji et al., 2013); 6. VOC `Actions` (Everingham et al.); 7. `Letters` (de Campos et al., 2009); and 8. `SVHN` (Netzer et al., 2011) datasets. We have rerun the different methods and obtain the same reported results as in Aljundi et al. (2017).

# B  EXTRA RESULTS

## B.1  PERMUTED MNIST SEQUENCE

In section 4.1, we have studied the performance of different regularizers and activation functions on 5 permuted Mnist tasks in a network with a hidden layer of size 128. Figure 5 shows the average accuracies achieved by each of the studied methods at the end of the learned sequence in a network with a hidden layer of size 64. Similar conclusions can be drawn. `Maxout` and `LWTA` perform similarly and improve slightly over `ReLU`. Regularizers applied to the representation are more powerful for sequential learning than regularizers applied directly to the parameters. Specifically, `L1-Rep` (orange) is consistently better than `L1-Param` (pink). Our `SLNID` is able of maintaining a good performance on all the tasks, achieving among the top average test accuracies. Admittedly, the performances of `SLNID` is very close to L1-Rep. The difference between these methods stands out more clearly for larger networks and more complex tasks.

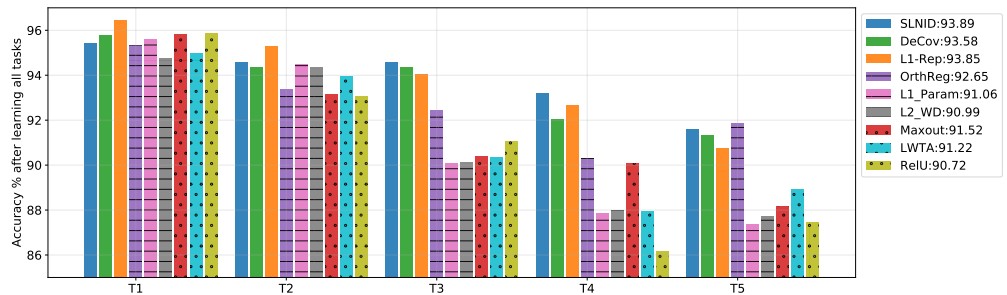

Figure 5: Comparison of different regularization techniques on 5 permuted MNIST sequence of tasks, hidden size=64. Representation based regularizers are solid bars, bars with lines represent parameters regularizers, dotted bars represent activation functions. See Figure 2 for size 128.

## B.2  SLNI WITH EWC

To show that our approach is not limited to `MAS` (Aljundi et al., 2017), we have also experimented with EWC (Kirkpatrick et al., 2016) as another importance weight based method along with our regularize `SLNID` on the permuted Mnist sequence. Figure 6 shows the test accuracy of each task at the end of the 5 permuted Mnist sequence achieved by our `SLNID` combined with `EWC` and by `No-Reg` (here indicating `EWC` without regularization). It is clear that `SLNID` succeeds to improve the performance on all the learned tasks which validates the utility of our approach with different sequential learning methods.

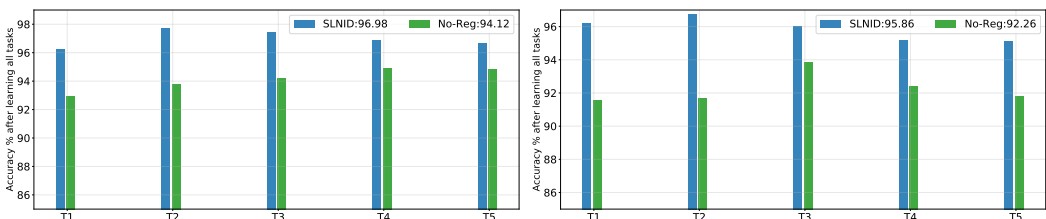

Figure 6: (a) `SLNID` with `EWC` on 5 permuted Mnist sequence of tasks, hidden size=128, (b) hidden size=64.

## B.3  CIFAR 100 SEQUENCE

In section 4.2 we have tested `SLNID` and other representation regularizers on the Cifar 100 sequence. In Figure 3(a) we compare their performance on a network with hidden layer size 256. Figure 7 repeats the same experiment for a network with hidden size 128. While `DeCov` and `SLNID` continue to improve over `No-Reg`, `L1-Rep` seems to suffer in this case. Our interpretation is that `L1-Rep` here interferes with the previously learned tasks while penalizing activations and hence suffers from catastrophic forgetting. In line with all the previous experiments `SLNID` achieves the best accuracies and manages here to improve over 6% compared to `No-Reg`.

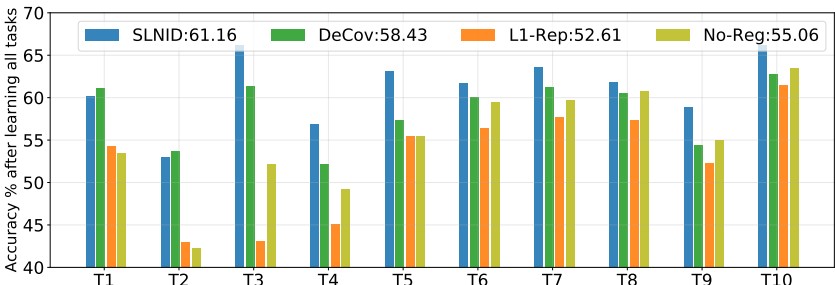

Figure 7: Comparison of different regularization techniques on a sequence of ten tasks from Cifar split. Hidden size=128. See Figure 3(a) for size 256.

## B.4 SPATIAL LOCALITY TEST

To avoid penalizing all the active neurons, our `SLNID` weights the correlation penalty between each two neurons based on their spatial distance using a Gaussian function. We want to visualize the effect of this spatial locality on the neurons activity. To achieve this, we have used the first 3 tasks of the Permuted Mnist sequence as a test case and visualized the neurons importance after each task. This is done using the network of hidden layer size 64. Figure 8, Figure 9 and Figure 10 show the neurons importance after each task. The left column is without locality, i.e. `SLNID`, and the right column is `SLNID`. Blue represents the first task, orange the second task and green the third task. When using `SLNID`, inhibition is applied in a local manner allowing more active neurons which could potentially improve the representation power. When learning the second task, new neurons become important regardless of their closeness to first task important neurons as those neurons are excluded from the inhibition. As such, new neurons are becoming active as new tasks are learned. For `SLNID` all neural correlation is penalized in the first task. And for later tasks, very few neurons are able to become active and important for the new task due to the strong global inhibition, where previous neurons that are excluded from the inhibition are easier to be re-used.

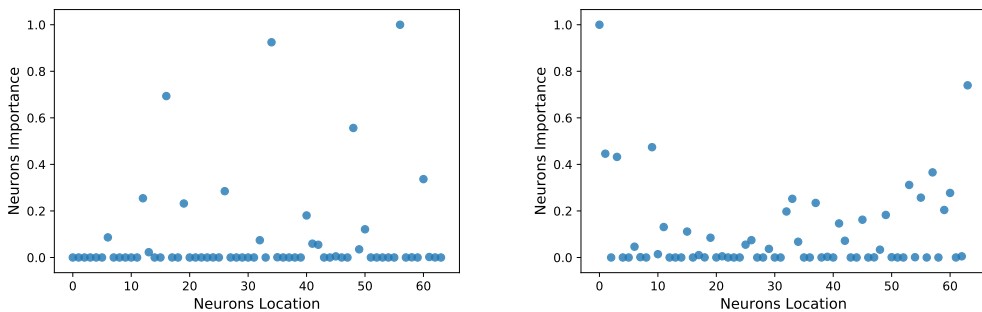

Figure 8: First layer neuron importance after learning the first task (blue). Left: `SNID`, Right: `SLNID`. More active neurons are tolerated in `SLNID`.

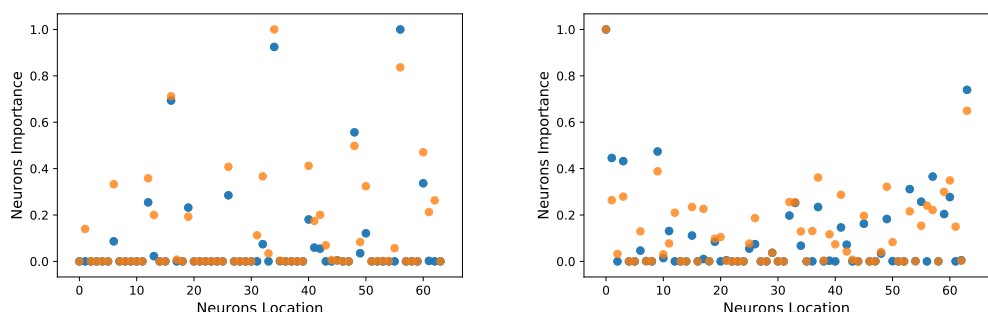

Figure 9: First layer neuron importance after learning the second task (orange), superimposed on Figure 8. Left: `SNID`, Right: `SLNID`. `SLNID` allows new neurons, especially those that were close neighbours to previous important neurons, to become active and to be used for the new task. `SNID` penalizes all unimportant neurons equally. As a result, previous neurons are adapted for the new tasks and less new neurons are getting activated.

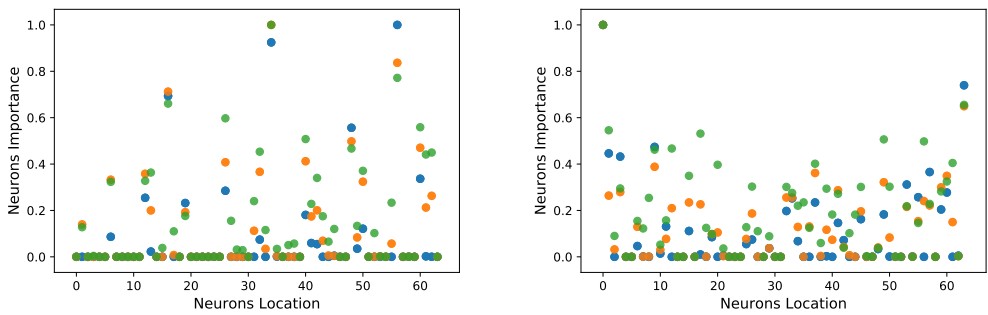

Figure 10: First layer neuron importance after learning the third task (green), superimposed on Figure 9. Left: `SNID`, Right: `SLNID`. `SLNID` allows previous neurons to be re-used for the third task. It avoids changing the previous important neurons by adding new neurons. For `SNID`, very few neurons are newly deployed. The new task is learned mostly by adapting previous important neurons, causing more interference.

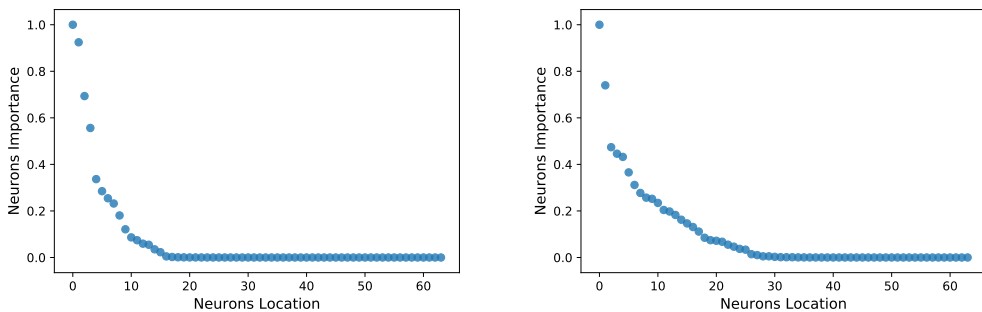

Figure 11: First layer neuron importance after learning the first task, sorted in descending order according to the first task neuron importance (blue). Left: SNID, Right: SLNID. More active neurons are tolerated in SLNID.

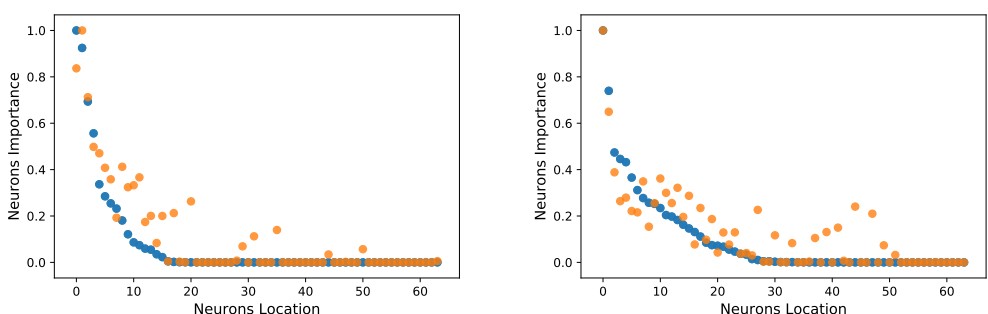

Figure 12: First layer neuron importance after learning the second task sorted in descending order according to the first task neuron importance (orange), superimposed on top of figure 11. Left: SNID, Right: SLNID. SLNID allows new neurons to become active and be used for the new task. SNID penalizes all unimportant neurons equally and hence more neurons are re-used then initiated for the first time.

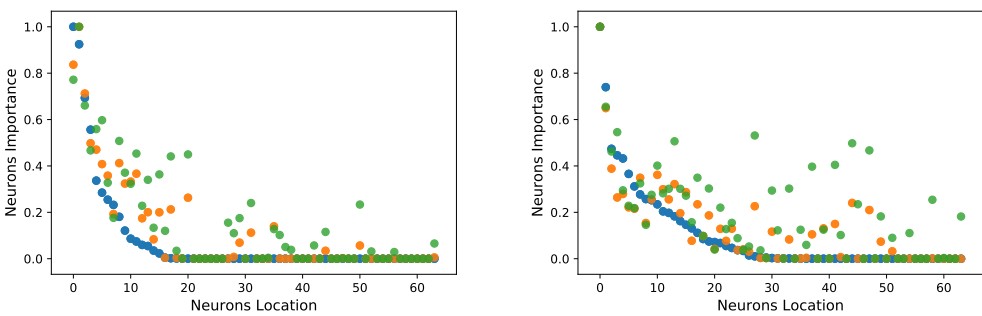

Figure 13: First layer neuron importance after learning the third task sorted in descending order according to the first task neuron importance (green), superimposed on top of figure 12. Left: SNID, Right: SLNID. SLNID allows previous neurons to be re-used for the third task while activating new neurons to cope with the needs of the new task. For SNID, very few neurons are newly deployed while most previous important neurons for previous tasks are re-adapted to learn the new task.

