# OpenReview forum: "Selfless Sequential Learning"
_ICLR.cc/2019/Conference_

### Official Review · AnonReviewer1 · 2018-11-01
**Thorough continual learning work but limited to task-based case**

**Rating:** 6
**Confidence:** 5

**Review:**

[REVISION]
The work is thorough and some of my minor concerns have been addressed, so I am increasing my score to 6. I cannot go beyond because of the incremental nature of the work, and the very limited applicability of the used continual learning setup from this paper.

[OLD REVIEW]
The paper proposes a novel, regularization based, approach to the sequential learning problem using a fixed size model. The main idea is to add extra terms to the loss encouraging representation sparsity and combating catastrophic forgetting. The approach fairs well compared to other regularization based approaches on MNIST and CIFAR-100 sequential learning variants.

Pros:
Thorough experiments, competitive baselines and informative ablation study.
Good performance on par or superior to baselines.
Clear paper, well written.

Cons:
The approach, while competitive in performance, does not seem to fix any significant issues with baseline methods. For example, task boundaries are still used, which limits applicability; in many scenarios which do have a continual learning problem there are no clear task boundaries, such as data distribution drift in both supervised and reinforcement learning.
Since models used in the work are very different from SOTA models on those particular tasks, it is hard to determine from the paper how the proposed method influences these models. In particular, it is not clear whether these changes to the loss would still allow top performance on regular classification tasks, e.g. CIFAR-10 or MNIST even without sequential learning, or in multitask learning settings.

Summary:
Although the work is substantial and experiments are thorough, I have reservations about extrapolating from the results to settings which do have a continual learning problem. Although I am convinced results are slightly superior to baselines, and I appreciate the lengthy amount of work which went into proving that, the paper does not go sufficiently beyond previous work.

---

> ### Author Response · Authors · 2018-11-22
> **No Task Boundaries exp and some clarificatoins**
>
> We thank AnonReviewer1 for their suggestions and comments.
> Note that we revised the paper, and renamed our full model to SLNID. Below are our comments to the main points:
>
> 1) Task boundaries are still used, which limits applicability; in many scenarios which do have a continual learning problem there are no clear task boundaries, such as data distribution drift in both supervised and reinforcement learning.
>
>  We agree with the reviewer on the importance of the mentioned setting where there are no clear task boundaries and distribution gradually drifts. Although this is orthogonal to the contribution of this work, we tested a setup where the data distribution drifts between tasks. When evaluating in this setting, we find, interestingly, that our proposed SNLID again works well in this setting compared to the LLL approach MAS (Aljundi et al.,2017), which benefits from hard task boundaries. Details can be found in the revised paper in Section 4.2 and Table 3. We believe this is an interesting setting to study further in future work.
>
> 2) Since models used in the work are very different from SOTA models on those particular tasks, it is hard to determine from the paper how the proposed method influences these models. In particular, it is not clear whether these changes to the loss would still allow top performance on regular classification tasks, e.g. CIFAR-10 or MNIST even without sequential learning, or in multitask learning settings.
>  Improving the state of the art results on non sequential scenarios is not the aim of this proposed regularizer. Further, the studied setup of LLL where data from previous or future task is not available during the training of a given task is much harder and challenging than joint training or multi task training where all data is available at training time.  In sec 4.2 Table 2 we compare against and outperform  state of the art LLL methods under the same setting and models used in those methods.

---

### Official Review · AnonReviewer2 · 2018-11-02
**Review for Selfless Sequential Learning**

**Rating:** 6
**Confidence:** 4

**Review:**

This paper deals with the problem of catastrophic forgetting in lifelong learning, which has recently attracted much attention from researchers. In particular, authors propose the regularized learning strategies where we are given a fixed network structure (without requiring additional memory increases in the event of new task arriving) in the sequential learning framework, without the access to datasets of previous tasks.  Performance comparisons were performed experimentally against diverse regularization methods including ones based on representation, based on parameter itself, and the superiority of representation-based regularization techniques was verified experimentally. Based on this, authors propose a regularization scheme utilizing the correlation between hidden nodes called SNI and its local version based on Gaussian weighting. Both regularizers are even extended to consider the importance of hidden nodes. Through MNIST, CIFAR, and tiny Imagenet datasets, it has been experimentally demonstrated that the proposed regularization technique outperforms state-of-the-art in sequential learning.

It is easy to follow (and I enjoyed the way of showing their final method, starting from SNI to SLNI and importance weighting). Also it is interesting that authors obtained meaningful results on several datasets beating state-of-the-arts based on very simple ideas.

However, given Cogswell et al. (2015) or Xiong et al. (2016), it seems novelty is somehow incremental (I could recognize that this work is different in the sense that it considers  local/importance based weighting as well as penalizing correlation based on L1 norm). Moreover, there is a lack of reasoning about why representation based regularization is more effective for life-long learning setting. Figure 1 is not that intuitive and it does not seem clearly describe the reasons.

My biggest concern with the proposed regularization technique is the importance of neurons in equation (6). It is doubtful whether the importance of activation of neurons based on "current data" is sufficiently verified in sequential learning (in the experimental section, avg performance for importance weight sometimes appears to come with performance improvements but not always). It would be great if authors can show some actual overlaps of activations across tasks (not just simple histogram as in Figure 5). And isn't g_i(x_m) a scalar? Explain why we need the norm when you get alpha.

It would be nice to clarify what the task sequence looks like in Figure 2. It is hard to understand that task 5, which is the most recent learning task, has the lowest performance in all tasks.

-----------------------------------------------------------------------------------------------------
- On figure 4: I knew histograms are given in figure 4 (I said figure 5 mistakenly, but I meant figure 4). But showing overlap patterns across tasks (at different layers for instance) might be more informative.
- On figure 2: It looks weird to me because last task has the lowest accuracy even for ReLU (sequential learning w/o regularization); tuning for task 5 will lead catastrophic forgetting for previous tasks, meaning acc for task 1 be the lowest?

-----------------------------------------------------------------------------------------------------
-  My concerns about figures are solved; I want to thank authors for their efforts.

---

> ### Author Response · Authors · 2018-11-22
> **Updated version, Neuron Importance exp., Figures showing overlap and clarifications**
>
> We thank AnonReviewer2 for their constructive comments, below is our reply to the main points. Note that we revised the paper, and renamed our full model to SLNID.
>
> 1) Reasoning about why representation based regularization is more effective for life-long learning setting.
>
> Please check our updated version.
>
> 2) Importance of neurons in equation (6)
>
> The importance of the neurons for a previous task is  computed based on that previous task data right after training that task. This is in line with estimating the importance weight in LLL methods. While on permuted mnist, the neurons importance doesn’t seem crucial to achieve the best performance, it improves the performance on Cifar, Tiny Imagenet, and the 8 Object recognition sequence. In fact,  permuted mnist is a simple scenario we use to compare all the studied methods  in a setting where the differences between tasks are easily identified. The full permutation requires the network to instantiate a new representation for the new task that associates new collections of pixels to digits patterns. In such a simple case, the importance of the neuron doesn’t seem a crucial factor while in more complicated sequence where tasks overlap and relatedness is much higher the neuron importance term is a key component. In Sec2.4 Table3 we again compare our regularizer with and without neurons importance,
> both  when evaluating the average performance using each task model and when using the last trained model.  While both SLNI with and without neurons importance improve the individual models accuracy (73.03 and 72.14 respectively), the performance at the end of the sequence (using the lastly trained model) significantly drops for SLNI without neurons importance (72.14 to 63.54) compared to SLNI with the neuron importance (73.03 to 70.75). This is a clear indication of the role of neurons importance in the sequential learning scenario in excluding previously important neurons from the penalty and hence avoiding interference between  tasks.
>
>  3) It would be great if authors can show some actual overlaps of activations across tasks (not just simple histogram as in Figure 5).
>
> Figure 4, bottom, shows the histogram of the mean activation on the first task achieved by each method.  Figures 8 & 9 & 10 in the Appendix show the neurons importance  after each task. It can be seen how new activations are initiated while reusing previous neurons. Also Figures 11&12&13, newly added, show the importance of the neurons sorted by the importance computed at the first task. It can be clearly seen which neurons are re-used and which are getting activated for the new task.
>
> 4) And isn't g_i(x_m) a scalar? Explain why we need the norm when you get alpha.
>
> In case of neurons in fully connected layers, g_i(x_m) is indeed a scalar. In the convolutional layers, importance of neurons is the norm of the gradient vector. While we only consider fully connected layers in this work, this was for sake of generality.
> Further, while estimating the importance, gradients are accumulated from different samples. We want to estimate how much a change in the previous task could happen when changing this neuron’s output. We are not interested in the sign of the change itself, hence we accumulate the absolute value of the gradients from different samples.
> For sake of clarity, we replaced the norm with absolute value sign in the new version.
>
> 5) It would be nice to clarify what the task sequence looks like in Figure 2. It is hard to understand that task 5, which is the most recent learning task, has the lowest performance in all tasks.
>
> In Figure 2, the sequence is first task mnist and remaining tasks permuted mnist with different permutations. Training individual models, results in similar accuracy in each task.  In the sequential setting, the last task is the most recent task. The model has to learn this task without forgetting all the previous tasks. As such, little capacity is left for the very last task. This is a known phenomenon in Lifelong learning and explained in Section 2 second paragraph. For this reason our regularizer always achieves the best performance on the last task in the sequence as it aims at leaving capacity for later tasks.  Also as mentioned in Section 4.1, Experimental Setup,  we have used a high value of (\lambda_omega) that ensures the least forgetting which allows us to test the effect on the later tasks performance.  For example, in the experiments Section 4.6, the accuracy on the last task for Finetuning is  90.0%  (as it forgets completely the previous tasks and only cares about the last task) while for MAS it is 68.2%. Our regularizer improves the accuracy on the last task to 77%, as more capacity is left for the last task. In the paper we only report the avg.acc at the end of the sequence due to  space limits.

---

> ### Author Response · Authors · 2018-11-27
> **Figure 4,2**
>
> On figure 4: I knew histograms are given in figure 4 (I said figure 5 mistakenly, but I meant figure 4). But showing overlap patterns across tasks (at different layers for instance) might be more informative.
>
> In figure 8&9&10, we have shown the important neurons in the first layer after learning each task, color coded with the task id. First task important neurons are in Blue, second task important neurons in orange and 3rd task important neurons in green. Figure 11 & 12&13 show the important neurons ordered with respect to their importance estimated at the first task. It can be seen from these figures how neurons are re-reused (overlapped) and other are newly activated for each new task. If the reviewer has other suggestions we will be eager to add.
>
> - On figure 2: It looks weird to me because last task has the lowest accuracy even for ReLU (sequential learning w/o regularization); tuning for task 5 will lead catastrophic forgetting for previous tasks, meaning acc for task 1 be the lowest?
>
>   All the baselines and compared methods in Figure 2 and 3 have importance weight regularizer (MAS), hence forgetting is minimized in all compared methods, accuracy in task 1 is preserved while scarifying accuracy in task 5. ReLU (sequential learning w/o regularization):  we main no additional sparsity regularizer. We thank the reviewer for pointing  this out, we have clarified  it in the revised version. Note that our regularizer improves 4-8% over No-Reg that uses already MAS as LLL method.
> We will be happy to clarify any other points.

---

> > ### Author Response · Authors · 2018-11-30
> > **Performance of Finetuning in Figure 2**
> >
> > To confirm the behavior of a network trained without LLL method (MAS) , we run the ReLU baseline without MAS on the sequence of 5 permuted mnist tasks and obtained the following accuracies at the end of the sequence:
> > Finetuning (ReLU NoMAS)= [40.79, 49.16, 72.5, 86.56, 97.08]
> > Compared to:
> > ReLU (with MAS)=[95.8, 93.66, 93.32, 89.95, 89.89]
> > SLNID( Ours)=[96.46, 96.25, 95.86, 95.81, 94.77]
> >
> > As the reviewer suggested Finetuning  (ReLU without MAS) has a better performance on the last task while forgetting severely the first task. However, as we mentioned earlier all our baselines in Figures 2 &3 run with MAS as a LLL method.

---

> ### Author Response · Authors · 2018-12-04
> **Thanks & comment on neuron importance**
>
> Thank you for checking the figures, kindly, we would like to draw your attention to the neuron importance experiment we newly added in Table 3. Accounting for neuron importance played a crucial role in reducing interference  and even when not using a LLL regularizer.

---

### Official Review · AnonReviewer3 · 2018-11-06
**Interesting work addressing an interesting class of problems, novel regularizer and good experiments, but writing and paper organization need work**

**Rating:** 7
**Confidence:** 4

**Review:**

REVISION AFTER REBUTTAL
While the revision does not address all of my concerns about clarity, it is much better. I still think that the introduction is overly long and the subsequent sections repeat information; if this were shortened there could be room for some of the figures that are currently in appendix. I appreciate the new figures; I think that it would be great if especially figure 10 were included in the main paper.
I agree with the other two reviewers that the work is somewhat incremental, but the differences are well explained, the experimental results are interesting (particularly the differences of parameter vs representation-based sparsity, and the plots in appendix showing neuron importance over tasks), and the progression from SNI to SLNID is well-presented.  I think overall that this paper is a good contribution and I recommend acceptance. I have updated my review to a 7.
===============
"Activations" "Representation" and "Outputs" are used somewhat interchangably throughout the work; for anyone not familiar it might be worth mentioning something about this in the intro.

Problem setting is similar to open set learning (classification); could be worth mentioning algorithms for this in the related work which attempt to set aside capacity for later tasks.

Results are presented and discussed in the introduction, and overall the intro is a bit long, resulting in parts later being repetitive.

Worth discussing sparsity vs. distributed representations in the intro, and how/where we want sparsity while still having a distributed representation.

Should be made clear that this is inspired by one kind of inhibition, and there are many others (i.e. inhibition in the brain is not always about penalizing neurons which are active at the same time, as far as I know)

Changes in verb tense throughout the paper make it hard to follow sometimes. Be consistent about explaining equations before or after presenting them, and make sure all terms in the equation are defined (e.g. SNI with a hat is used before definition). Improper or useless "However" or "On the other hand" to start a lot of sentences.

Figure captions could use a lot more experimental insight and explanation - e.g. what am I supposed to take away from Figure 10 (in appendix B4), other than that the importance seems pretty sparse? It looks to me like there is a lot of overlap in which neurons are important or which tasks, which seems like the opposite of what the regularizer was trying to achieve. This is a somewhat important point to me; I think this interesting and I'm glad you show it, but it seems to contradict the aim of the regularizer.

How does multi-task joint training differ from "normal" classification? The accuracies especially for CIFAR seem very low.

Quality: 7/10 interesting and thoughtful proposed regularizer and experiments; I would be happy to increase this rating if the insights from experiments, especially in the appendix, are a bit better explained
Clarity:  6/10 things are mostly clearly explained although frequently repetitive, making them seem more confusing than they are. If the paper is reorganized and the writing cleaned up I would be happy to increase my rating because I think the work is good.
Originality: 8/10 to my knowledge the proposed regularizer is novel, and I think think identifying the approach of "selfless" sequential learning is valuable (although I don't like the name)
Significance: 7/10 I am biased because I'm interested in LLL, but I think these problems should receive more attention.

Pros:
 - proposed regularizer is well-explained and seems to work well, ablation study is helpful

Cons:
 - the intro section is almost completely repetitive of section 3 and could be significantly shortened, and make more room for some of the experimental results to be moved from the appendix to main text
 - some wording choices and wordiness make some sentences unclear, and overall the organization and writing could use some work

Specific comments / nits: (in reading order)
1. I think the name "selfless sequential learning" is a bit misleading and sounds like something to do with multiagent cooperative RL; I think "forethinking" or something like that that is an actual word would be better, but I can't think of a good word... maybe frugal?
2.  Mention continual/lifelong learning in the abstract
3. "penalize changes" maybe "reduce changes" would be better?
4. "in analogy to parameter importance" cite and explain parameter importance
5. "advocate to focus on selfless SL" focus what? For everyone doing continual learning to focus on methods which achieve that through leaving capacity for later tasks? This seems like one potentially good approach, but I can imagine other good ones (e.g. having a task model)
6. LLL for lifelong learning is defined near the end of the intro, should be at the beginning when first mentioned
7. "lies at the heart of lifelong learning" I would say it is an "approach to lifelong learning"
8. "fixed model capacity" worth being specific that you mean (I assume) fixed architecture and number of parameters
9. "those parameters by new tasks" cite this at the end of the sentence, otherwise it is unclear what explanation goes with which citation
10.  "hard attention masks, and stored in an embedding" unclear what is stored in the embedding. It would be more helpful to explain how this method relates to yours rather than just describing what they do.
11. I find the hat notation unclear; I think it would be better just to have acronyms for each setting and write out the acronyms in the caption
12."richer representation is needed and few active neurons can be tolerated" should this be "more active neurons"?
13. Comparison with state of the art section is repetitive of the results sections

---

> ### Author Response · Authors · 2018-11-22
> **We have updated our paper taking into account the suggested edits and  here are some clarifications**
>
> We thank AnonReviewer3 for their constructive comments, below is our reply to the main points.
> Note that we revised the paper, and renamed our full model to SLNID.
>
> 1) Changing the hat notation:
>
> Following the suggestion, we adapted the naming as follows an used them throughout the paper. Note that  SLNID now corresponds to the complete version of our regularizer:
> - Sparse coding through Neural Inhibition (SNI)
> - Sparse coding through Local Neural Inhibition (SLNI)
> - Sparse coding through Local Neural Inhibition and Discounting (SLNID)
>
> 2) Results are presented and discussed in the introduction, and overall the intro is a bit long, resulting in parts later being repetitive.
>
> Please check our updated version.
>
> 3) Worth discussing sparsity vs. distributed representations in the intro, and how/where we want sparsity while still having a distributed representation.
>
> Please check our updated version.
>
> 4) Figure captions could use a lot more experimental insight and explanation - e.g.  Figure 10 (in appendix B4)
> We have updated the figures captions accordingly.
>
> From Figure  8 & 9 & 10  we can deduce two main points:
> - The important neurons are sparse, SLNID tolerates more active neurons than SNID.
> - With each new task,  new neurons are getting used and become important (Figure 9 & 10) .
> Figures 11&12&13, newly added,  where neurons are sorted w.r.t. their importance for the first task, show how new neurons are becoming important for the new tasks.
> Previous important neurons are also reused for the new tasks. This is not against our regularizer. Our regularizer avoids inhibiting neurons from previous tasks by excluding them, so they can be used freely (Equation 7, section 3.3). The LLL regularizer (Equation 1) ensures that their connections are not being changed drastically and hence performance preserved in previous tasks. So, both, achieving sparsity to leave space for future tasks and sharing important neurons, whenever possible, allowing forward transfer, are actually goals of our regularizer.
>
> 5) How does multi-task joint training differ from "normal" classification? The accuracies especially for CIFAR seem very low.
>
> The shown performance of joint training represents the average accuracy achieved on each task by masking out  classifier scores of the other tasks when computing the arg max. However, the training was done using a shared 100-dimensional classification layer. We use a small network with only 128 or 256 neurons in the hidden layer, training it for 50 epochs with SGD optimizer and a learning rate of 0.01. No dropout was used, no batch normalization and no data augmentation.  Our aim was to set a fair comparison between different regularizers without the interference of other mechanisms. We did not aim for state of the art results on learning jointly a dataset.

---

> ### Author Response · Authors · 2018-12-02
> **Thanks**
>
> Thank you so much, we will try to shorten and move the figures in a final version.

---

### Meta-Review · Area_Chair1 · 2018-12-15
**incremental but interesting contribution to life long learning for neural networks.**

**Confidence:** 4
**Recommendation:** Accept (Poster)

**Metareview:**

Two of the reviewers raised their scores during the discussion phase noting that the revised version was clearer and addressed some of their concerns.  As a result, all the reviewers ultimately recommended acceptance.  They particularly enjoyed the insights that the authors shared from their experiments and appreciated that the experiments were quite thorough.  All the reviewers mentioned that the work seemed somewhat incremental, but given the results, insights and empirical evaluation decided that it would still be a valuable contribution to the conference.  One reviewer added feedback about how to improve the writing and clarity of the paper for the camera ready version.